# GeoAI: A Model-Agnostic Meta-Ensemble Zero-Shot Learning Method for Hyperspectral Image Analysis and Classification

**Konstantinos Demertzis * and Lazaros Iliadis**

Department of Civil Engineering, School of Engineering, Democritus University of Thrace, Xanthi 67100, Greece; liliadis@civil.duth.gr

**\*** Correspondence: kdemertz@fmenr.duth.gr; Tel.: +30-694-824-1881

**Abstract:** Deep learning architectures are the most effective methods for analyzing and classifying Ultra-Spectral Images (USI). However, effective training of a Deep Learning (DL) gradient classifier aiming to achieve high classification accuracy, is extremely costly and time-consuming. It requires huge datasets with hundreds or thousands of labeled specimens from expert scientists. This research exploits the MAML++ algorithm in order to introduce the *Model-Agnostic Meta-Ensemble Zero-shot Learning* (MAME-ZsL) approach. The MAME-ZsL overcomes the above difficulties, and it can be used as a powerful model to perform Hyperspectral Image Analysis (HIA). It is a novel optimization-based Meta-Ensemble Learning architecture, following a *Zero-shot Learning* (ZsL) prototype. To the best of our knowledge it is introduced to the literature for the first time. It facilitates learning of specialized techniques for the extraction of user-mediated representations, in complex Deep Learning architectures. Moreover, it leverages the use of first and second-order derivatives as pre-training methods. It enhances learning of features which do not cause issues of exploding or diminishing gradients; thus, it avoids potential overfitting. Moreover, it significantly reduces computational cost and training time, and it offers an improved training stability, high generalization performance and remarkable classification accuracy.

**Keywords:** model-agnostic meta-learning; ensemble learning; GIS; hyperspectral images; deep learning; remote sensing; scene classification; geospatial data; Zero-shot Learning

## 1. Introduction

Hyperspectral image analysis and classification is a timely special field of Geoinformatics which has attracted much attention recently. This has led to the development of a wide variety of new approaches, exploiting both spatial and spectral content of images, in order to optimally classify them into discrete components related to specific standards. Typical information products obtained by the above approaches are related to diverse areas; namely: ground cover maps for environmental Remote Sensing; surface mineral maps used in geological applications; vegetation species maps, employed in agricultural-geoscience studies and in urban mapping. Recent developments in optical sensor technology and Geoinformatics (GINF), provide multispectral, Hyperspectral (HyS) and panchromatic images at very high spatial resolution. Accurate and effective HyS image analysis and classification is one of the key applications which can enable the development of new decision support systems. They can provide significant opportunities for business, science and engineering in particular. Automatic assignment of a specific semantic label to each object of a HyS image (according to its content) is one of the most difficult problems of GINF Remote Sensing (RES).

With the available HyS resolution, subtle objects and materials can be extracted by HyS imaging sensors with very narrow diagnostic spectral bands. This can be achieved for a variety of purposes, such as detection, urban planning, agriculture, identification, surveillance and quantification. HyS image analysis enables the characterization of objects of interest (e.g., land cover classes) with unprecedented accuracy, and keeps inventories up to date. Improvements in spectral resolution have called for advances in signal processing and exploitation algorithms.

A Hyperspectral image is a 3D data cube, which contains two-dimensional spatial information (image feature) and one-dimensional spectral information (spectral bands). Especially, the spectral bands occupy very fine wavelengths. Additionally, image features related to land cover and shape disclose the disparity and association among adjacent pixels from different directions at a confident wavelength. This is due to its vital applications in the design and management of soil resources, precision farming, complex ecosystem/habitat monitoring, biodiversity conservation, disaster logging, traffic control and urban mapping.

It is well known that increasing data dimensionality and high redundancy between features might cause problems during data analysis. There are many significant challenges which need to be addressed when performing HyS image classification. Primarily, supervised classification faces a challenge related to the imbalance between high dimensionality and incomplete accessibility of training samples, or to the presence of mixed pixels in the data. Further, it is desirable to integrate the essential spatial and spectral information, so as to combine the complementary features which stem from source images.

Deep Learning methodologies have significantly contributed towards the evolution and development of HyS image analysis and classification [1]. Deep Learning (DL) is a branch of computational intelligence which uses a series of algorithms that model high-level abstraction data using a multi-level processing architecture.

It is difficult for all Deep Learning algorithms to achieve satisfactory classification results with limited labeled samples, despite their undoubtedly well-established functions and their advantages. The approaches with the highest classification accuracy and generalization ability fall under the supervised learning umbrella. For this reason, especially in the case of Ultra-Spectral Images, huge datasets with hundreds or thousands of specimens labeled by experts are required [2]. This process is very expensive and time consuming.

In the case of supervised image classification, the input image is processed by a series of operations performed at different neuronal levels. Eventually, the output generates a probability distribution for all possible classes (usually using the *Softmax* function). Softmax is a function which takes an input vector Z of *k* real numbers and normalizes it into a probability distribution consisting of *k* probabilities, proportional to the exponentials of the input numbers [3].

$$\sigma(z)_j = \frac{e^{z_j}}{\sum_{k=1}^{K} e^{z_k}} j = 1, \dots, k \ where \ \sigma : R^k \to R^k \ Z = (z_1, \dots \dots, z_k) \in R^k \tag{1}$$

For example, if we try to classify an image as $L_{im-a}$, or $L_{im-b}$, or $L_{im-c}$, or $L_{im-d}$, then we generate four probabilities for each input image, indicating the respective probabilities of the image belonging to each of the four categories. There are two important points to be mentioned here. First, during the training process, we require a large number of images for each class ($L_{im-a}$, $L_{im-b}$, $L_{im-c}$, $L_{im-d}$). Secondly, if the network is only trained for the above four image classes, then we cannot expect to test it for any other class; e.g., "$L_{im-x}$." If we want our model to sort images as well, then we need to get many "$L_{im-x}$" images and to rebuild and retrain the model [3]. There are cases where we do not have enough data for each category, or the classes are huge but also dynamically changing. Thus, the cost of data collection and periodic retraining is enormous. A reliable solution should be sought in these cases. In contrast, *k*-shot learning is a framework within which the network is called upon to learn quickly and with few examples. During training, a limited number of examples from diverse classes with their labels are introduced. The network is required to learn general characteristics of the problem, such as features which are either common to the samples of the same class, or unique features which differentiate and eventually separate the classes.

In contrast to the learning process of the traditional neural networks, it is not sufficient for the network to learn good representations of the training classes, as the testing classes are distinct and they are not presented in training. However, it is desirable to learn features which distinguish the existing classes.

The evaluation process consists of two distinct stages of the following format [4]:

Step 1: Given $k$ examples (value of $k$-shot), if $k = 1$, then the process is called *one-shot*; if $k = 5$, *five-shot*, and so on. The parameter $k$ represents the number of labeled samples given to the algorithm by each class. By considering these samples, which comprise the support set, the network is required to classify and eventually adapt to existing classes.

Step 2: Unknown examples of the labeled classes are presented randomly, unlike the ones presented in the previous step, which the network is called to correctly classify. The set of examples in this stage is known as the query set.

The above procedure (steps) is repeated many times using random classes and examples which are sampled from the testing-evaluation set.

As it is immediately apparent from the description of the evaluation process, as the number of classes increases, the task becomes more difficult, because the network has to decide between several alternatives. This means that *Zero-shot Learning* [5] is clearly more difficult than the *one-shot*, which is more difficult than the *five-shot*, and so on. Although humans have the ability to cope with this process, traditional ANN require many more examples to generalize effectively, in order to achieve the same degree of performance. The limitation of these learning approaches is that the model has access to minimum samples from each class and the validation process is performed by calculating the cross-entropy error of the test set. Specifically, in the cases of one-shot and *Zero-shot Learning* (ZsL), only one example each of the candidate classes and only meta-data is shown at the evaluation stage.

Overall, *k-shot learning* is a perfect example of a problematic area, where specialized solutions are needed to design and train systems capable to learn very quickly from a small support set, containing only 1–5 samples per class. These systems can offer strong generalization to a corresponding target set. A successful exploitation of the above k-shot learning cases is provided by meta-learning techniques which can be used to deliver effective solutions [6].

In this work, we propose a new classification model, which is based on zero-shot philosophy, named MAME-ZsL. The significant advantages of the proposed algorithm is that it reduces computational cost and training time; it avoids potential overfitting by enhancing the learning of features which do not cause issues of exploding or diminishing gradients; and it offers an improved training stability, high generalization performance and remarkable classification accuracy. The superiority of the proposed model refers to the fact that the instances in the testing set belong to classes which were not contained in the training set. In contrast, the traditional supervised state-of-the-art Deep Learning models were trained with labeled instances from all classes. The performance of the proposed model was evaluated against state-of-the-art supervised Deep Learning models. The presented numerical experiments provide convincing arguments regarding the classification efficiency of the proposed model.

## 2. Meta-Learning

It is a field of machine learning where advanced learning algorithms are applied to the data and metadata of a given problem. The models "*learn to learn*" [7] from previous learning processes or previous sorting tasks they have completed [8]. It is an advanced form of learning where computational models, which usually consist of multiple levels of abstraction, can improve their learning ability. This is achieved by learning some or all of their own building blocks, through the experience gained in handling a large number of tasks. Their building blocks which are "*learning to learn*" can be optimizers, loss functions, initializations, Learning Rates, updated functions and architectures.

In general, for real physical modeling situations, the input patterns with and without tags are derived from the same boundary distribution or they follow a common cluster structure. Thus, classified data can contribute to the learning process, while correspondingly useful information

related to the exploration of the data structure of the general set can be extracted from the non-classified data. This information can be combined with knowledge originating from prior learning processes or from completed prior classification tasks. Based on the above theory, *meta-learning* techniques can discover the structure of the data, by allowing new tasks to be learned quickly. This is achieved by using different types of metadata, such as the properties of the learning problem, the properties of the algorithm used (e.g., performance measures) or the patterns derived from data from a previous problem. This process employs knowledge from unknown cases sampled from real-world distribution of examples, aiming to enhance the outcome of the learning task. In this way it is possible to learn, select, change or combine different learning algorithms to effectively solve a given problem.

*Meta-learning* is achieved by conceptually dividing learning in two levels. The inner-most levels acquire specific knowledge for specific tasks (e.g., fine-tuning a model on a new dataset), while the outer-most levels acquire across-task knowledge (e.g., learning to transfer tasks more efficiently).

If the inner-most levels are using learnable parameters, outer-most optimization process can meta-learn the parameters of such components, thereby enabling automatic learning of inner-loop components.

A *meta-learning* system should combine the following three requirements [9,10]:

1.  The system must include a learning sub-system.
2.  Experience must be derived from the use of extracted knowledge from metadata related to the dataset under consideration, or from previous learning tasks completed in similar or different fields.
3.  Learning biases should be dynamically selected.

Employing a generic approach, a credible meta-learning model should be trained in a variety of learning tasks and it should be optimized for the best performance in generalizing tasks, including potentially unknown ones. Each task is associated with a dataset *D*, containing attribute vectors and class labels in a supervised learning problem. The optimal parameters of the model are [11]:

$$\theta^* = arg_\theta^{min} \mathbb{E}_{D \sim P(D)}[L_\theta(D)] \tag{2}$$

It looks similar to a normal learning process, but each data set is considered as a data sample.

The dataset *D* is divided in two parts, a training set *S* and a set of predictions *B* for validation and testing.

$$D = \langle S, B \rangle \tag{3}$$

The dataset *D* includes pairs of vectors and labels so which:

$$D = \{(x_i, y_i)\} \tag{4}$$

Each label belongs to a known label set *L*.

Let us consider a classifier $f_\theta$. The parameter $\theta$ extracts the probability $x, P_\theta(y|x)$ of a data point to belong to class $y$, given by the attribute vector. Optimal parameters should maximize the likelihood of identifying true labels in multiple training batches $B \subset D$:

$$\theta^* = argmax_\theta \mathbb{E}_{(x,y) \in D}[P_\theta(y|x)] \tag{5}$$

$$\theta^* = argmax_\theta \mathbb{E}_{B \subset D}\left[ \sum_{(x,y) \in B} P_\theta(y|x) \right] \tag{6}$$

The aim is to reduce the prediction error in data samples with unknown labels, in which there is a small set of "fast learning" support which can be used for "fine-tuning".

Fast learning is a trick which creates a "fake" dataset containing a small subset of labels (to avoid exposing all labels to the model). During the optimization process, various modifications take place, aiming to achieve rapid learning.

A brief step-by-step description of the whole process is presented below [11]:

1.  Development of a subset of labels $L_s \subset L$.

2. Development of a training subset $S^L \subset D$ and a forecast subset $B^L \subset D$. Both of them include data points with labels belonging to the subset $L_s$, $y \in L_s$, $\forall (x, y) \in S^L$, $B^L$.

3. The optimization procedure uses $B^L$ to calculate the error and to update the model parameters via back propagation. This is done in the same way as it is used in a simple supervised learning model.

In this way each sample pair $(S^L, B^L)$ can be considered to be a data point. The model is trained so that it can generalize to new unknown datasets.

The following function (Equation (7)) is a modification of the supervised learning model. The symbols of the meta-learning process have been added:

$$\theta^* = argmax_\theta \mathbb{E}_{L_s \subset L} \left[ \mathbb{E}_{S^L \subset D, B^L \subset D} \left[ \sum_{(x,y) \in B^L} P_\theta(x, y, S^L) \right] \right] \tag{7}$$

There are three *meta-learning* modeling approaches, as presented below [11] and the Table 1.:

a. Model-based: These are techniques based on the use of circular networks with external or internal memory. They update their parameters quickly with minimal training steps. This can be achieved through their internal architecture or by using other control models. *Memory-augmented neural networks* and *meta networks* are characteristic cases of *model-based meta-learning* techniques.

b. Metrics-based: These are techniques based on learning effective distance measurements which can offer generalization. The core concept of their operation is similar to that of the nearest neighbors algorithms, where they aim to learn a measurement or a distance from objects. The concept of a good metric depends on the problem, as it should *represent* the relationship between inputs to the site, facilitating problem solving. *Convolutional Siamese neural networks, matching networks, relation networks and prototypical networks* are characteristic metrics-based meta-learning techniques.

c. Optimization-based: These are techniques based on optimizing the parameters of the model for quick learning. *LSTM Meta-Learners, temporal discreteness and the reptile plus Model-Agnostic Meta-Learning* (MAML) algorithms are typical cases of optimization-based meta-Learning.

**Table 1.** Meta-learning approaches.

|  | **Model-Based** | **Metric-Based** | **Optimization-Based** |
|---|---|---|---|
| Key idea | RNN; memory | Metric learning | Gradient descent |
| How is $P_\theta(y\|x)$ modeled? | $f_\theta(x, S)$ | $\sum_{(x_i, y_i) \in S} k_\theta(x, x_i) y_i$ | $P_{g_\varphi(\theta, S^L)}(y\|x)$ |

\* $k_\theta$ is a kernel function which calculates the similarity between $x_i$ and $x$.

The *Recurrent Neural Networks* (RNNs) which use only internal memory, and also the *Long-Short-Term Memory* approaches (LSTM), are not considered meta-learning techniques [11]. Meta-learning can be achieved through a variety of learning examples. In this case, the *supervised gradient-based* learning can be considered as the most effective method [11]. More specifically, the *gradient-based, end-to-end differentiable meta-learning*, provides a wide framework for the application of effective *meta-learning* techniques.

This research proposes an *optimization-based, gradient-based, end-to-end* differentiable meta-learning architecture, based on an innovative evolution of the MAML algorithm [10]. MAML is one of the most successful and at the same time simple optimization algorithms which belongs to the meta-learning approach. One of its great advantages is that it is compatible with any model which learns through the Gradient Descent (GRD) method. It is comprised of the Base-Learner (BL) and the Meta-Learner (ML) models, with the second used to train the first. The weights of the BL are updated following the GRD method in learning tasks of the k-shot problem, whereas the ML applies the GRD approach on the weights of the BL, before the GRD [10].

In Figure 1 you can see a depiction of the MAML algorithm.

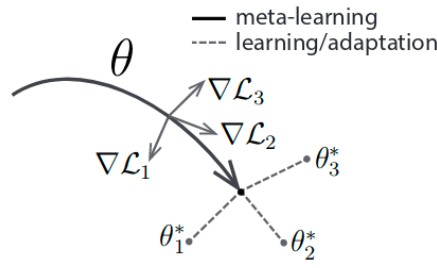

**Figure 1.** Model-Agnostic Meta-Learning algorithm.

It should be clarified that $\theta$ denotes the weights of the meta-learner. Gradient $L_i$ comprises of the losses for task $i$ in a meta-batch. The $\theta_i *$ are the optimal weights for each task. It is essentially an optimization procedure on a set of parameters, such that when a slope step is obtained with respect to a particular task $i$, the respective parameters $\theta_i^*$ are approaching their optimal values. Therefore, the goal of this approach is to learn an intrinsic feature which is widely applicable to all tasks of a distribution $p(T)$ and not to a single one. This is achieved by minimizing the total loss across tasks sampled from the distribution $p(T)$.

In particular, we have a base-model represented by a parametric function $f_\theta$ with parameters $\theta_i$ and a task $T_i \sim p(T)$. After applying the Gradient Descent, a new feature vector is obtained denoted as $\theta_i'$:

$$\theta_i' = \theta - \alpha \nabla_\theta L_{T_i}(f_\theta) \tag{8}$$

We will consider that we execute only one GD step. The meta-learner optimizes the new parameters using the initial ones, based on the performance of the $f_{\theta'}$ model, in tasks which use sampling from the $P(T)$. Equation (9) is the meta-objective [10]:

$$\min_\theta \sum_{T_i \sim p(T)} L_{T_i}\left(f_{\theta_i'}\right) = \min_\theta \sum_{T_i \sim p(T)} L_{T_i}\left(f_\theta - \alpha \nabla_\theta L_{T_i}(f_\theta)\right) \tag{9}$$

The meta-optimization is performed again with the *Stochastic Gradient Descent* (SGD) and it updates the parameters $\theta$ as follows:

$$\theta \leftarrow \theta - \beta \nabla_\theta \sum_{T_i \sim p(T)} L_{T_i}\left(f_{\theta_i'}\right) \tag{10}$$

It should be noted that we do not actually define an additional set of variables $\theta_i'$ whose values are calculated by considering one (or more) Gradient Descents from $\theta$ relative to *task i*. This step is known as the *Inner Loop Learning (INLL)*, which is the reverse process of the *Outer Loop Learning (OLL)*, and it optimizes Equation (10). If, for example, we apply INLL to fine-tune $\theta$ for process $i$, then according to Equation (10) we are optimizing a target with the expectation which the model applies to each task, following corresponding fine-tuning procedures.

The following Algorithm 1 is an analytical presentation of the MAML algorithm [10].

---

**Algorithm 1.** MAML.

---

**Require:** $p(T)$: distribution over tasks
**Require:** $\alpha, \beta$: step size hyperparameters
1: randomly initialize $\theta$
2: **while** not done **do**
3: Sample batch of tasks $T_i \sim p(T)$
4: **for all** $T_i$ **do**
5: Evalluate $\nabla_\theta L_{T_i}(f_\theta)$ with respect to K examples
6: Compute adapted parameters with gradient descent: $\theta_i' = \theta - \alpha \nabla_\theta L_{T_i}(f_\theta)$
7: **end for**
8: Update $\theta \leftarrow \theta - \beta \nabla_\theta \sum_{T_i \sim p(T)} L_{T_i}\left(f_{\theta_i'}\right)$
9: **end while**

---

Figure 2 is a graphical illustration of the operation of the MAML algorithm.

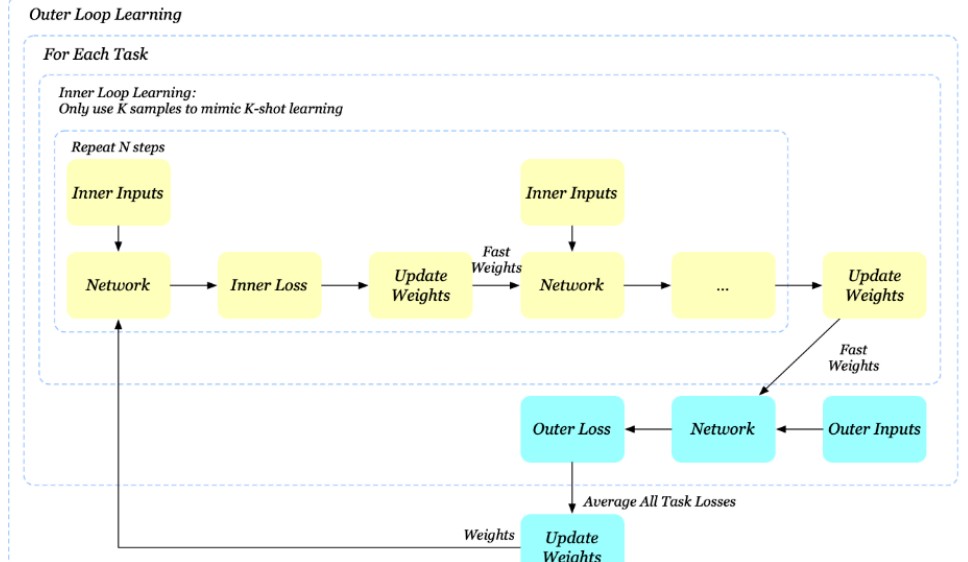

**Figure 2.** Graphical presentation of the MAML.

It should be clarified that the intermediate parameters $\theta_i$' are considered fast weights. The INLL considers all of the $N$ gradient steps for the final estimation of the fast weights, based on the fact that the outer learning loop calculates the outer task loss $L_{T_i}\left(f_{\theta_i'}\right)$. However, though the inner learning loop makes $N$ iterations, the MAML algorithm employs only the final weights to perform the OLL. However, this is a fairly significant problem, as it can create instability in learning when $N$ is large.

The field of few-shot or Zero-shot Learning, has recently seen substantial advancements. Most of these advancements came from casting few-shot learning as a meta-learning problem. MAML is currently one of the best approaches for few-shot learning via meta-learning. It is a simple, general, and effective optimization algorithm that does not place any constraints on the model architecture or loss functions. As a result, it can be combined with arbitrary networks and different types of loss functions, which makes it applicable to a variety of different learning processes. However, it has a variety of issues, such as being very sensitive to neural network architectures, often leading to instability during training. It requires arduous hyperparameter searches to stabilize training and achieve high generalization, and it is very computationally expensive at both training and inference times.

## 3. Related Work

Although the MAML algorithm and its variants do not use parameters other than those of the base-learner, network training is quite slow and computationally expensive as it contains second-degree derivatives. In particular, the meta-update of the MAML algorithm includes gradient nested in gradient, or second-degree derivatives, which significantly increases the computational cost. In order to solve the above problem, several approximation techniques have been proposed to accelerate the algorithm.

Finn et al. [12] developed the MAML by ignoring the second derivatives, calculating the slope in the meta-update, which they called FOMAML (First Order MAML).

More specifically, MAML optimizes the:

$$\min_{\theta}\mathbb{E}_{T\sim p(T)}\left[L_T\left(\mathbb{U}_T^k(\theta)\right)\right], \tag{11}$$

where $\mathbb{U}_T^k$ is the process by which k samples are taken from task $T$ and $\theta$ is updated. This procedure employs the support set and the query set, so the optimization can be rewritten as follows:

$$\min_{\theta}\mathbb{E}_{T\sim p(T)}\left[L_{T,Q}\left(\mathbb{U}_{T,S}(\theta)\right)\right] \tag{12}$$

Finally, MAML uses the slope method to calculate the following:

$$gMAML = L_{T,Q}\left(\mathbb{U}_{T,S}(\theta)\right) = \mathbb{U}'_{T,S}(\theta)L'_{T,Q}(\tilde{\theta}), \tag{13}$$

where $\tilde{\theta} = \mathbb{U}_{T,S}(\theta)$ and $\mathbb{U}'_{T,S}$ is the Jacobian renewal matrix of $\mathbb{U}_{T,S}$ where the FOMAML considers $\mathbb{U}'_{T,S}$ as unitary, so it calculates the following:

$$gFOMAML = L'_{T,Q}(\tilde{\theta}) \tag{14}$$

The resulting method still calculates the meta-gradient for the parameter values after updating $\theta'$, which is an effective post-learning method from a theoretical point of view. However, experiments have shown that the yield of this method is almost the same as the one obtained by a second derivative. Most of the improvement in MAML comes from the gradients of the objective at the post-update parameter values, rather than the second-order updates from differentiating through the gradient update.

A different implementation employing first degree derivatives was studied and analyzed by Nichol et al. [13]. They introduced the reptile algorithm, which is a variation of the MAML, using only the first derivative. The basic difference from FOMAML is that the last step treats $\theta^{\sim}\theta$ as a slope and feeds it into an adaptive algorithm such as ADAM. Algorithm 2 presents reptile.

---

**Algorithm 2.** Reptile algorithm.

---

Initialize $\theta$, the vector of initial parameters
1: **for** $iteration = 1,2, ...,$ **do**
2: Sample task $T$, corresponding to Loss $L_T$ on weighs vector $\tilde{\theta}$
3: Compute $\tilde{\theta} = \mathbb{U}_T^k(\theta)$, denoting k steps of gradient descent or Adam algorithm
4: Update $\theta \leftarrow \theta + \varepsilon(\tilde{\theta} - \theta)$
5: **end for**

---

MAML also suffers from training instability, which can currently only be alleviated by arduous architecture and hyperparameter searches.

Antoniou et al. proposed an improved variant of the algorithm, called MAML++, which effectively addresses MAML problems, by providing a much improved training stability and removing the dependency of training stability on the model's architecture. Specifically, Antoniou et al. [14] found that simply replacing max-pooling layers with stridden convolutional layers makes network training unstable. It is clearly shown in Figure 3 that in two of the three cases, the original MAML appears to be unstable and irregular, while all 3 MAML++ models appear to converge consistently very quickly, with much higher generalization accuracy without any stability problems.

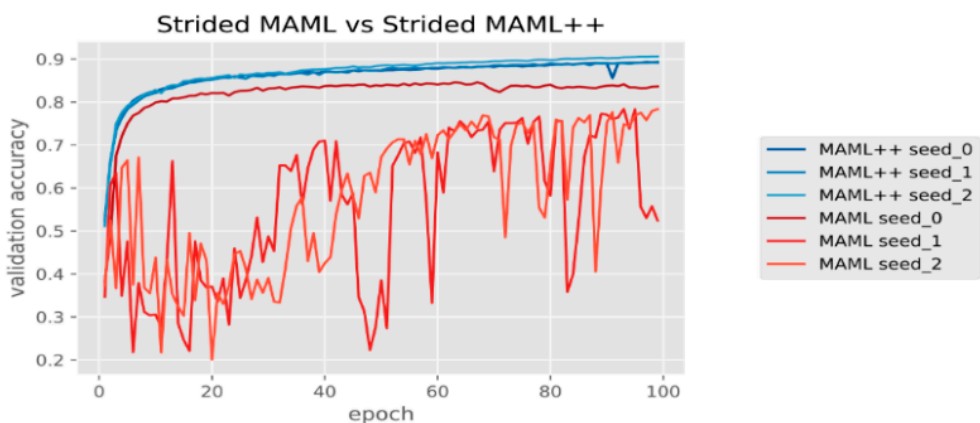

**Figure 3.** Stabilizing MAML.

It was estimated that the instability was caused by a gradient degradation (gradient explosion or vanishing gradient) which was due to the depth of the network. Let us consider a typical four-layer Convolutional Neural Network (CNN) followed by a single linear layer. If we repeat the Inner Loop Learning N times, then the inference graph comprises 5N layers in total, without any skip connections.

Since the original MAML only uses the final weights for the Outer Loop Learning, the backpropagation algorithm has to go through all layers, which causes gradient degradation. To solve the above problem, the Multi-Step Loss (MSL) optimization approach was adopted. It eliminates the problem by calculating the external loss after each internal step, based on the outer loop update, as in Equation (15) below:

$$\theta = \theta - \beta \nabla_\theta \sum_{i=1}^{B} \sum_{j=1}^{N} w_j L_{T_i}\left(f_{\theta_j^i}\right), \tag{15}$$

where $\beta$ is a Learning Rate; $L_{T_i}\left(f_{\theta_j^i}\right)$ denotes the outer loss of task $i$ when using the base-network weights after the $j$-inner-step update; and $w_j$ denotes the importance weight of the outer loss at step $j$.

The following, Figure 4, is a graphical display of the MAML++ algorithm, where the outer loss is calculated after each internal step and the weighted average is obtained at the end of the process.

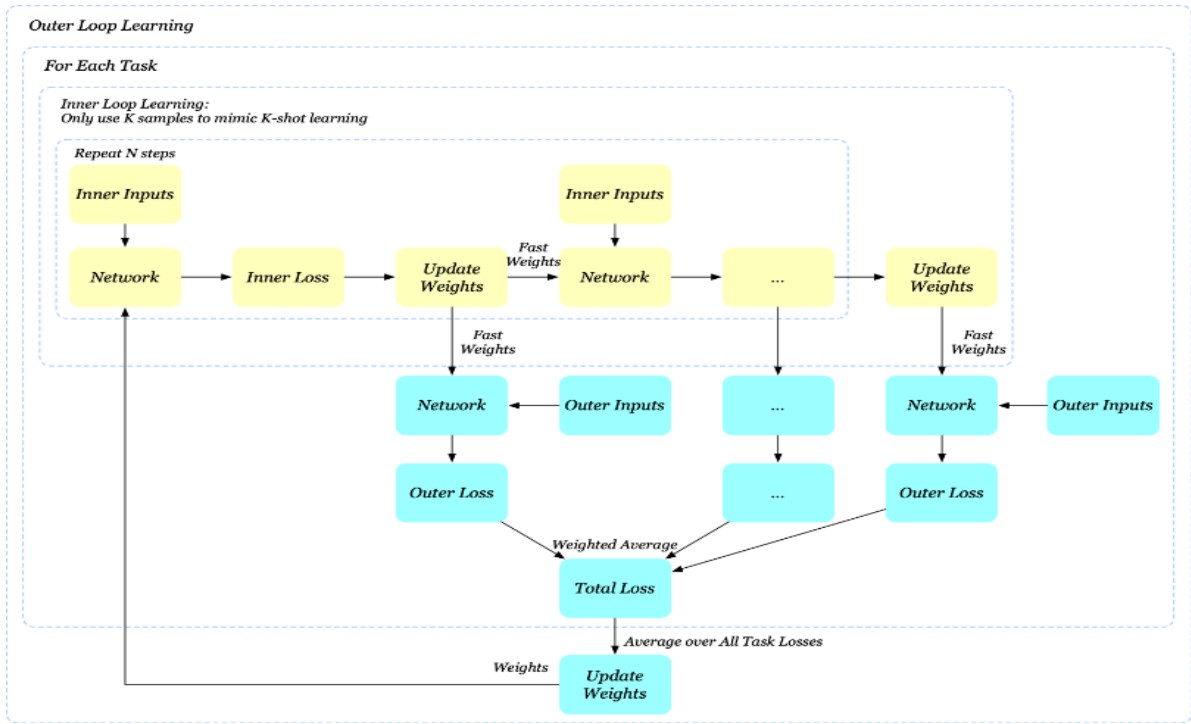

**Figure 4.** MAML++ visualization.

In practice, all losses are initialized with equal contributions to the overall loss, but as repetitions increase, contributions from previous steps are reduced, while the ones of subsequent steps keep increasing gradually. This is to ensure that as training progresses, the final step loss receives more attention from the optimizer, thereby ensuring that the lowest possible loss is achieved. If the annealing is not used, the final loss might be higher compared to the one obtained by the original formulation. Additionally, due to the fact that the original MAML overcomes the second-order derivative cost by completely ignoring it, the final general performance of the network is reduced. The MAML++ solves this problem, by using the *derivative order annealing* method. Specifically, it employs the first order grade for the first 50 epochs of training and then it moves to the second order grade for the rest of the training process. An interesting observation is that this derivative-order

annealing does not create incidents of exploding or diminishing gradients, and so the training is much more stable.

Another drawback of MAML is the fact that it does non-use *batch normalization statistic accumulation*. Instead, only the statistics of the current batch are used. As a result, smoothing is less effective, as the trained parameters must include a variety of different means and standard deviations from different tasks. A naive application would accumulate current batch statistics at all stages of the Inner Loop Learning update, which could cause optimization problems, and it could slow or stop optimization altogether. The problem stems from the erroneous assumption that the original model and all its updated iterations have similar attribute distributions. Thus, the current statistics could be shared across all updates to the internal loop of the network. Obviously, this assumption is not correct. A better alternative solution, which is employed by MAML++, is the storage of *per-step batch normalization statistics* and the reading of *per-step batch normalization weights and* biases for each repetition of the inner loop. One issue that affects the speed of generalization and convergence is the use of a *shared Learning Rate* for all parameters and all steps of learning process update.

The consistent Learning Rate requires multiple hyperparameter searches, in order to find the right rate for a particular set of data. This is computationally expensive and time consuming, depending on how the search is shared. Use the shared Learning Rate for all parameters and all the steps of updating the learning process. In addition, while gradient is an effective data fitting tactic, a constant Learning Rate can easily lead to overfitting under the *few-shot regime*. An approach to avoid potential overfitting is the identification of all learning factors in a way that maximizes the power of generalization rather than the over-fitting of the data.

Li et al. [15] proposed a Learning Rate for each parameter in the core network where the internal loop was updated, as in the following equation (Equation (16)):

$$\theta' = \theta - \alpha \circ \nabla_\theta L_{T_i}(f_\theta), \tag{16}$$

where $\alpha$ is a vector of learnable parameters with the same size as $L_{T_i}(f_\theta)$ and $\circ$ denotes the element-wise product operation. We do not put the constraint of positivity on the Learning Rate (LER) denoted as "$\alpha$." Therefore, we should not expect the inner-update direction to follow the gradient direction.

A clearly improved approach to the above process is suggested by MAML++ which employs *per-layer per-step* Learning Rates. For example, if it is assumed that the *core network* comprises L layers and the *Inner Loop Learning* consists of N stages of updating, then there are LN additional learnable parameters for the *Inner Loop Learning Rate*.

MAML uses the *ADAM* algorithm with a constant LER to optimize the meta-objective. This means which more time is required to properly adjust the Learning Rate, which is a critical parameter of the generalization performance. On the other hand, MAML++ employs the cosine annealing scheduling on the *meta-optimizer*, which is defined based on the following Equation (17) [16].

$$\beta = \beta_{min} + \frac{1}{2}(\beta_{max} - \beta_{min})\left(1 + \cos\left(\frac{T}{T_{max}}\pi\right)\right), \tag{17}$$

where $\beta_{min}$ denotes the minimum Learning Rate, $\beta_{max}$ denotes the initial Learning Rate, $T$ is the number of current iterations and $T_{max}$ is the maximum number of iterations. When $T = 0$, the LER $\beta = \beta_{max}$. On the other hand, if $T = T_{max}$, then $\beta = \beta_{min}$. In practice, we might want $T$ to be $T_{max}$.

In summary, this particular MAML++ standardization enables its use in complex Deep Learning architectures, making it easier to learn more complex functions, such as loss functions, optimizers or even gradient computation functions. Moreover, the use of first-class derivatives offers a powerful pre-training method aiming to detect the parameters which are less likely to cause exploding or diminishing gradients. Finally, the *learning per-layer per-step* LER technique avoids potential overfitting, while it significantly reduces the computational cost and time required to build a consistent Learning Rate throughout the process.

## 4. Design Principles and Novelties of the Introduced MAME-ZsL Algorithm

As it has already been mentioned, the proposed MAME-ZsL algorithm employs MAML++ for the development of a robust *Hyperspectral Image Analysis* and *Classification* (HIAC) model, based on ZsL. The basic novelty introduced by the improved MAME-ZsL model, is related to a neural network with Convolutional (CON) filters, comprising very small receptive fields of size 3 × 3.

The Convolutional stride and the spatial padding were set to 1 pixel. Max-pooling was performed over 3 × 3 pixels windows with stride equal to three. All of the CON layers were developed using the Rectified Linear Unit (ReLU) nonlinear Activation Function (ACF), except for the last layer where the Softmax ACF [3] was applied, as it performs better on multi-classification problems like the one under consideration (18).

$$\sigma_j(z) \quad = \frac{e^{z_j}}{\sum_{k=1}^{K} e^{z_k}}, j = 1, \dots, K \tag{18}$$

The Sigmoid approach offers better results in binary classification tasks. It is a fact that in the Softmax, the sum of probabilities is equal to 1, which is not the case for the Sigmoid. Moreover, in Softmax the highest value has a higher probability than the others, while in the Sigmoid the highest value is expected to have a high probability but not the highest one.

The fully Convolutional Neural Network (CNN) was trained based on the novel *AdaBound* algorithm [17] which employs dynamic bounds on the Learning Rate and it achieves a smooth transition to stochastic gradient descent. Algorithm 3 makes a detailed presentation of the AdaBound [17]:

---

**Algorithm 3.** The AdaBound algorithm.

---

**Input:** $x_1 \in F$, initial step size $\alpha, \{\beta_{1t}\}_{t=1}^{T}, \beta_2$ lower bound function $\eta_l$, upper bound function $\eta_u$

1: Set $m_0 = 0, u_0 = 0$
2: **for** $t = 1$ **to** T **do**
3: $g_t = \nabla f_t(x_t)$
4: $m_t = \beta_{1t} m_{t-1} + (1 - \beta_{1t}) g_t$
5: $u_t = \beta_2 u_{t-1} + (1 - \beta_2) g_t^2$ and $V_t = diag(u_t)$
6: $\hat{\eta}_t = Clip\left(\frac{a}{\sqrt{V_t}}, \eta_l(t), \eta_u(t)\right)$ and $\eta_t = \frac{\hat{\eta}_t}{\sqrt{t}}$
7: $x_{t+1} = \prod_{f, diag(\eta_t^{-1})}(x_t - \eta_t \cdot m_t)$
8: **end for**

---

Compared to other methods, AdaBound has two major advantages. It is uncertain whether there exists a fixed turning point to distinguish the simple ADAM algorithm and the SGD. The advantage of the AdaBound is the fact that it addresses this problem with a continuous transforming procedure, rather than with a "hard" switch. The AdaBound introduces an extra hyperparameter to perform the switching time, which is not very easy to fine-tune. Moreover, it has a higher convergence speed than the stochastic gradient descent ones. Finally, it overcomes the poor generalization ability of the adaptive models, as it uses dynamic limits on the LER, aiming towards higher classification accuracy.

The selection of the appropriate hyperparameters to be employed in the proposed method, was based on the restrictions' settings and configurations, which should be based on the consideration of the different decision boundaries of the classification problem. For example, the obvious choice of classifiers with the smallest error in training data is considered as improper for generating a classification model. The performance based on a training dataset, even when cross-validation is used, may be misleading when first seen data vectors are used. In order for the proposed process to be effective, individual hyperparameters were chosen. They not only display a certain level of diversity, but they also use different operating functions, thus allowing different decision boundaries to be created and combined in such a way that can reduce the overall error.

In general, the selection of features was based on a heuristic method which considers the way the proposed method faces each situation. For instance:

- Are any parametric approaches employed?
- What is the effect of the outliers? (The use of a subgroup of training sets with bagging can provide significant help towards the reduction of the effect of outliers or extreme values)
- How is the noise handled? For instance: if it is repeatedly non-linear, it can detect linear or non-linear dispersed data; it tends to perform very well with a lot of data vectors. The final decision is made based on the performance encountered by the statistical trial and error method.

## 5. Application of the MAME-ZsL in Hyperspectral Image Analysis

Advances in artificial intelligence, combined with the extended availability of high quality data and advances in both hardware and software, have led to serious developments in the efficient processing of data related to the GeoAI field (*Artificial Intelligence and Geography/Geographic Information Systems*). Hyperspectral Image Analysis for efficient and accurate object detection using Deep Learning is one of the timely topics of GeoAI. The most recent research examples include detection of soil characteristics [18], detailed ways of capturing densely populated areas [19], extracting information from scanned historical maps [20], semantic point sorting [21], innovative spatial interpolation methods [22] and traffic forecasting [23].

Similarly, modern applications of artificial vision and imaging (IMG) systems significantly extend the distinctive ability of optical systems, both in terms of spectral sensitivity and resolution. Thus, it is possible to identify and differentiate spectral and spatial regions, which although having the same color appearance, are characterized by different physico-chemical and/or structural properties. This differentiation is based on the emerging spatial diversification, which is detected by observing in narrow spectral bands, within or outside the visible spectrum.

Recent technological developments have made it possible to combine IMG (spatial variation in RGB resolution) and spectroscopy (spectral analysis in spatially emitted radiation) in a new field called "Spectral Imaging" (SIM). In the SIM process, the intensity of light is recorded simultaneously as a function of both wavelength and position. The dataset corresponding to the observed surface contains a complete image, different for each wavelength. In the field of spectroscopy, a fully resolved spectrum can be recorded for each pixel of the spatial resolution of the observation field. The multitude of spectral regions, which the IMG system can manage, determines the difference between multispectral (tens of regions) and Hyperspectral (hundreds of regions) Imaging. The key element of a typical spectral IMG system is the monochromatic image sensor (monochrome camera), which can be used to select the desired observation wavelength.

It can be easily perceived that the success of a sophisticated Hyperspectral Analysis System (HAS) is a major challenge for DL technologies, which are using a series of algorithms attempting to model data characterized by high-level of abstraction. HAS use a multi-level processing architecture, which is based on sequential linear and non-linear transformations. Despite their undoubtedly well-established and effective approaches and their advantages, these architectures depend on the performance of training with huge datasets which include multiple representations of images of the same class. Considering the multitude of classes which may be included in a Hyperspectral image, we realize that this process is so incredibly time consuming and costly, that it can sometimes be impossible to run [1].

The ZsL method was adopted based on a heuristic [24], hierarchical parameter search methodology [25]. It is part of a family of learning techniques which exploit data representations to interpret and derive the optimal result. This methodology uses distributed representation, the basic premise of which is that the observed data result from the interactions of factors which are organized in layers. A fundamental principle is that these layers correspond to levels of abstraction or composition based on their quantity and size.

Fine-Grained Recognition (FIG_RC) is the task of distinguishing between visually very similar objects, such as identifying the species of a bird, the breed of a dog or the model of an aircraft. On the other hand, FIG_RC [26] which aims to identify the type of an object among a large number of subcategories, is an emerging application with the increasing resolution which exposes new details in image data. Traditional fully supervised algorithms fail to handle this problem where there is low

between-class variance and high within-class variance for the classes of interest with small sample sizes. The experiments show that the proposed fine-grained object recognition model achieves only 14.3% recognition accuracy for the classes with no training examples. This is slightly better than a random guess accuracy of 6.3%. Another method [27] automatically creates a training dataset from a single degraded image and trains a denoising network without any clear images. However, the performance of the proposed method shows the same performance as the optimization-based method at high noise levels.

Hu et al., 2015, proposed a time-consuming and resource depended model [28] which learns to perform zero-shot classification, using a meta-learner which is trained to produce corrections to the output of a previously trained learner. The model consists of a Task Module (TM) which supplies an initial prediction, and a Correction Module (CM) updating the initial prediction. The TM is the learner and the CM is the meta-learner. The correction module is trained in an episodic approach, whereby many different task modules are trained on various subsets of the total training data, with the rest being used as unseen data for the CM. The correction module takes as input a representation of the TM's training data to perform the predicted correction. The correction module is trained to update the task module's prediction to be closer to the target value.

In addition [29] proposes the use of the visual space as the embedding one. In this space, the subsequent nearest neighbor search suffers much less from the harness problem and it becomes more effective. This model design also provides a natural mechanism for multiple semantic modalities (e.g., attributes and sentence descriptions) to be fused and optimized jointly in an end-to-end manner. Only the statistics of the current environment are used and the trained process must include a variety of different statistics from different tasks and environments.

Additionally, [30] propose a very promising approach with high-grade accuracy, but the model is characterized by high bias. In the case of image classification, various spatial information can be extracted and used, such as edges, shapes and associated color areas. As they are organized into multiple levels, they are hierarchically separated into levels of abstraction, creating the conditions for selecting the most appropriate features for the training process. Utilizing the above processes, ZsL inspires and simulates the functions of human visual perception, where multiple functional levels and intermediate representations are developed, from capturing an image to the retina to responding in stimuli. This function is based on the conversion of the input representation to a higher level one, as it is performed by each intermediate unit. High-level features are more general and unchanged, while low-level ones help to categorize inputs. Their effectiveness is interpreted on the basis of the "*universal approximation theorem,*" which deals with the ability of a neural structure to approach continuous functions and the probabilistic inference which considers the activation of nonlinearity as a function of cumulative distribution. This is related to the concepts of optimization and generalization respectively [25].

Given that in deep neural networks, each hidden level trains a distinct set of features, coming from the output of the previous level, further operation of this network enables the analysis of the most complex features, as they are reconstructed and decomposed from layer to layer. This hierarchy, as well as the degradation of information, while increasing the complexity of the system, also enables the handling of high-dimensional data, which pass through non-linear functions. It is thus possible to discover unstructured data and to reveal a latent structure in unmarked data. This is done in order to handle more general problematic structures, even discerning the minimal similarities or anomalies they entail.

Specifically, since the aim was the design of a system with zero samples from the target class, the proposed methodology used the intermediate representations extracted from the rest of the image samples. This was done in order to find the appropriate representations to be used in order to classify the unknown image samples.

To increase the efficiency of the method, bootstrap sampling was used, in order to train different subsets of the data set in the most appropriate way. Bootstrap sampling is the process of using increasingly larger random samples until the accuracy of the neural network is improved. Each sample is used to compile a separate model and the results of each model are aggregated with

"voting"; that is, for each input vector, each classifier predicts the output variable, and finally, the value with the most "votes" is selected as the response variable for which particular vector. This methodology, which belongs to the ensemble methods, is called bagging and has many advantages, such as reducing co-variance and avoiding overfitting, as you can see in the below Figure 5 [31].

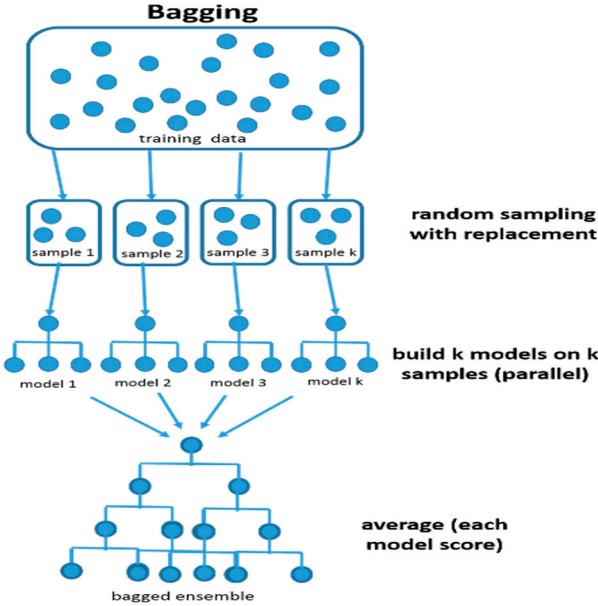

**Figure 5.** Bagging (bootstrap sampling) (https://www.kdnuggets.com/).

The ensemble approach was selected to be employed in this research, due to the particularly high complexity of the examined ZsL, and due to the fact that the prediction results were highly volatile. This can be attributed to the sensitivity of the correlational models to the data, and to the complex relationship which describes them. The ensemble function of the proposed system offers a more stable model with better prediction results. This is due to the fact that the overall behavior of a multiple model is less noisy than a corresponding single one. This reduces the overall risk of a particularly bad choice.

It is important to note that in Deep Learning, the training process is based on analyzing large amounts of data. The research and development of neural networks is flourishing thanks to recent advancements in computational power, the discovery of new algorithms and the increase in labeled data.

Neural networks typically take longer to run, as an increase in the number of features or columns in the dataset also increases the number of hidden layers. Specifically, we should say that a single affine layer of a neural network without any non-linearities/activations is practically the same as a linear model. Here we are referring to deep neural networks that have multiple layers and activation functions (non-linearities as Relu, Elu, tanh, Sigmoid) Additionally, all of the nonlinearities and multiple layers introduce a nonconvex and usually rather complex error space, which means that we have many local minimums that the training of the deep neural network can converge to. This means that a lot of hyperparameters have to be tuned in order to get to a place in the error space where the error is small enough so that the model will be useful. A lot of hyper parameters which could start from 10 and reach up to 40 or 50 are dealt with via Bayesian optimization, using Gaussian processes to optimize them, which still does not guarantee good performance. Their training is very slow, and adding the tuning of the hyperparameters into that makes it even slower, whereas the linear model would be much faster to be trained. This introduces a serious cost–benefit tradeoff. A trained linear model has weights which are interpretable and gives useful information to the data scientist as to how various features should have roles for the task at hand.

Modern frameworks like TensorFlow or Theano perform execution of neural networks on GPU. They take advantage of parallel programming capabilities for large array multiplications, which are typical of backpropagation algorithms.

The proposed Deep Learning model is a quite resource-demanding technology. It requires powerful, high-performance graphics processing units and large amounts of storage to train the models. Furthermore, this technology needs more time to train in comparison with traditional machine learning. Another important disadvantage of any Deep Learning model is that it is incapable of providing arguments about why it has reached a certain conclusion. Unlike in the case of traditional machine learning, you cannot follow an algorithm to find out why your system has decided which it is a tree on a picture, not a tile. To correct errors in Deep Learning, you have to revise the whole algorithm.

## 6. Description of the Datasets

The datasets used in this research include images taken from a Reflective Optics System Imaging Spectrometer (ROSIS). More specifically, the Pavia University and Pavia Center datasets were considered [32]. Both datasets came from the ROSIS sensor during a flight campaign over Pavia in southern Italy. The number of spectral bands is 102 for the Pavia Center and it is 103 for Pavia University. The selected Pavia Center and Pavia University images have an analysis of 1096 × 1096 pixels and 610 × 610 pixels respectively. Ultrasound imaging consists of 115 spectral channels ranging from 430 to 860 nm, of which only 102 were used in this research, as 13 were removed due to noise. Rejected specimens which in both cases contain no information (including black bars) can be seen in the following figure (Figure 6) below.

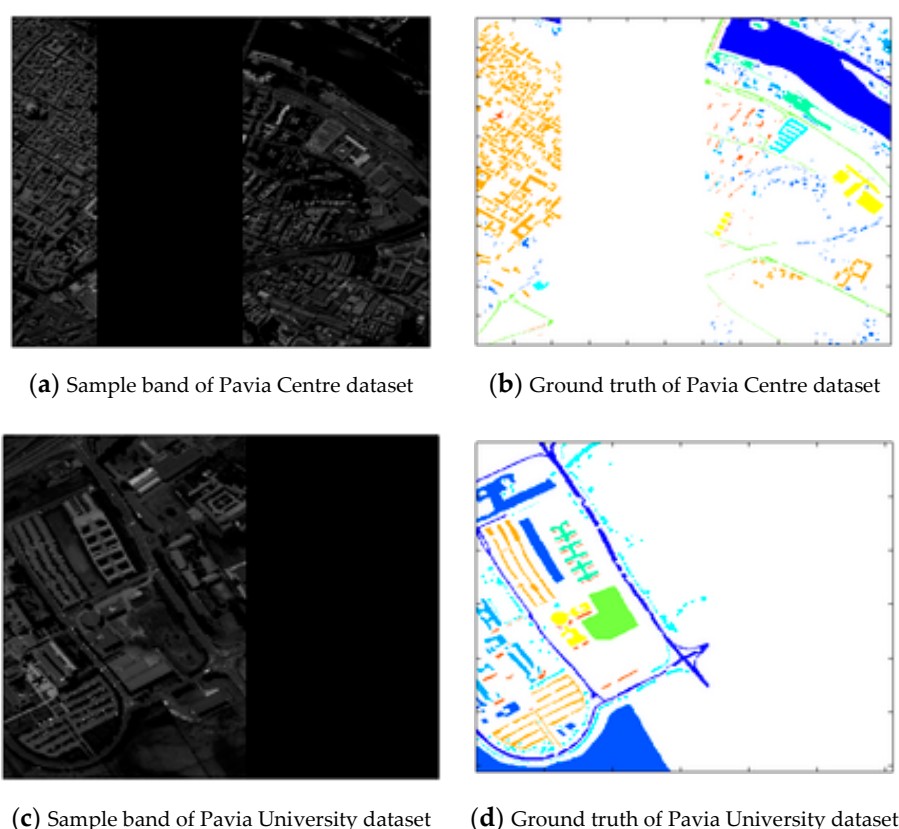

(**a**) Sample band of Pavia Centre dataset      (**b**) Ground truth of Pavia Centre dataset

(**c**) Sample band of Pavia University dataset      (**d**) Ground truth of Pavia University dataset

**Figure 6.** Noisy bands in Pavia Centre and University datasets.

The available samples were scaled down, so that every image has an analysis of 610 × 610 pixels and geometric analysis of 1.3 m. In both datasets the basic points of the image belong to nine categories which are mainly related to land cover objects. The Pavia University dataset includes nine

classes, and in total, 46,697 cases. The Pavia Center dataset comprises nine classes with 7456 cases, whereas the first seven classes are common in both datasets (*Asphalt, Meadows, Trees, Bare Soil, Self-Blocking Bricks, Bitumen, Shadows*) [33].

The Pavia University dataset was divided into training, validation and testing sets, as is presented in the following table (Table 2) [32].

**Table 2.** Pavia University dataset.

| Class No | Class Name | All Instances | Sets | | |
|---|---|---|---|---|---|
| | | | Training | Validation | Test |
| 1 | Asphalt | 7179 | √ | - | - |
| 2 | Meadows | 19,189 | √ | - | - |
| 3 | Trees | 3588 | √ | - | - |
| 4 | Bare Soil | 5561 | √ | - | - |
| 5 | Self-Blocking Bricks | 4196 | √ | - | - |
| 6 | Bitumen | 1705 | √ | - | - |
| 7 | Shadows | 1178 | √ | - | - |
| 8 | Metal Sheets | 1610 | - | √ | - |
| 9 | Gravel | 2491 | - | - | √ |
| | **Total** | **46,697** | **42,596** | **1610** | **2491** |

The Pavia Center dataset was divided into training, validation and testing sets, as is presented analytically in the following table (Table 3) [32].

**Table 3.** Pavia Center dataset.

| Class No | Class Name | All Instances | Sets | | |
|---|---|---|---|---|---|
| | | | Training | Validation | Test |
| 1 | Asphalt | 816 | √ | - | - |
| 2 | Meadows | 824 | √ | - | - |
| 3 | Trees | 820 | √ | - | - |
| 4 | Bare Soil | 820 | √ | - | - |
| 5 | Self-Blocking Bricks | 808 | √ | - | - |
| 6 | Bitumen | 808 | √ | - | - |
| 7 | Shadows | 476 | √ | - | - |
| 8 | Water | 824 | - | √ | - |
| 9 | Tiles | 1260 | - | - | √ |
| | **Total** | **7456** | **5372** | **824** | **1260** |

*Metrics Used for the Assessment of the Modeling Effort*

The following metrics were used for the assessment of the modeling effort [33,34]:

(a) Overall Accuracy (OA): This index represents the number of samples correctly classified, divided by the number of testing samples.

(b) Kappa Statistic: This is a statistical measure which provides information on the level of agreement between the truth map and the final classification map. It is the percentage of agreement corrected by the level of agreement, which could be expected to occur by chance. In general, it is considered to be a more robust index than a simple percent agreement calculation, since $k$ takes into account the agreement occurring by chance. It is a popular measure for benchmarking classification accuracy under class imbalance. It is used in static classification scenarios and for streaming data classification. Cohen's kappa measures the agreement between two raters, where each classifies $N$ items into $C$ mutually exclusive categories. The definition of $\kappa$ is [35,36]:

$$\kappa = \frac{p_0 - p_e}{1 - p_e} = 1 - \frac{1 - p_0}{1 - p_e}, \tag{19}$$

where $p_0$ is the relative observed agreement among raters (identical to accuracy), and $p_e$ is the hypothetical probability of chance agreement. The observed data are used to calculate the probabilities of each observer, to randomly see each category. If the raters are in complete agreement, then $\kappa = 1$. If there is no agreement among the raters other than what would be expected by chance (as given by $p_e$), then $\kappa \approx 0$.

The Kappa Reliability (KR) can be considered as the outcome from the data editing, allowing the conservancy of more relevant data for the upcoming forecast. A detailed analysis of the KR is presented in the following Table 4.

**Table 4.** Kappa Reliability.

| Kappa | Reliability |
|---|---|
| 0.00 | no reliability |
| 0.1–0.2 | minimum |
| 0.21–0.40 | little |
| 0.41–0.60 | moderate |
| 0.61–0.80 | important |
| ≥0.81 | maximum |

(c) McNemar test: The McNemar statistical test was employed to evaluate the importance of classification accuracy derived from different approaches [31]:

$$z_{12} = \frac{f_{12} - f_{21}}{\sqrt{f_{12} + f_{21}}},$$
(20)

where $f_{ij}$ is the number of correctly classified samples in classification $i$, and incorrect one are in classification $j$. McNemar's test is based on the standardized normal test statistic, and therefore the null hypothesis, which is "no significant difference," rejected at the widely used $p = 0.05$ ($|z| > 1.96$) level of significance.

## 7. Results and Comparisons

The training of the models was performed using a Learning Rate of 0.001. Loss function is the cross-entropy error between the predicted and true class. The cross-entropy error was used as the Loss function between the predicted and the true class. For the other parameters of the model, the recommended default settings were set as in [37].

A comparison with the following most widely used supervised Deep Learning models was performed to validate the effectiveness of the proposed architecture.

(a)   1-D CNN: The architecture of the 1-D CNN was designed as in [38], and it comprises the input layer, the Convolutional Layer (COL), the max-pooling layer, the fully connected layer and the output one. The number of the Convolutional Filters (COFs) was equal to 20, the length of each filer was 11 and the pooling size had the value of 3. Finally, 100 hidden units were contained in the fully connected layer.

(b)   2-D CNN: The 2-D CNN architecture was designed as in [39]. It includes three COLs which were supplied with 4×4, 5×5 and 4×4 CON filters respectively. The COL except of the final one were followed by max-pooling layers. Moreover, the numbers of COFs for the COLs were to 32, 64 and 128, respectively.

(c)   Simple Convolutional/deconvolutional network with the simple Convolutional blocks and the unpooling function, as it is described in [40,41].

(d)   Residual Convolutional/deconvolutional network: Its architecture used residual blocks and a more accurate unpooling function, as it is shown in [42].

The following Tables 5 and 6 show the classification maps which have emerged for the cases of the Pavia University and Pavia Center datasets. Moreover, they present a comparative analysis of the accuracy of the proposed MAME-ZsL algorithm, with the performance of the following classifiers; namely: 1-D CNN, 2-D CNN, Simple Convolutional/Deconvolutional Network (SC/DN), Residual Convolutional/Deconvolutional Network (RC/DN).

A single dataset including all records of the Pavia University and Pavia Center datasets was developed, which was employed in an effort to fully investigate the predictive capacity of the proposed system. It was named *General Pavia Dataset (GPD)*, and it was divided into training (Classes 1–7), validation (Classes 8 and 9) and testing subsets (Classes 10 and 11). It is presented in the following Table 7.

**Table 5.** Testing classification accuracy and performance metrics.

| Pavia University Dataset | | | | | | | | | | | | | | | |
|---|---|---|---|---|---|---|---|---|---|---|---|---|---|---|---|
| **Class Name** | **1-D CNN** | | | **2-D CNN** | | | **SC/DN** | | | **RC/DN** | | | **MAME-ZsL** | | |
| | **OA** | **κ** | **McN** | **OA** | **κ** | **McN** | **OA** | **κ** | **McN** | **OA** | **κ** | **McN** | **OA** | **κ** | **McN** |
| Metal Sheets | 99.41% | 0.8985 | | 100% | 1 | | 97.55% | 0.8356 | | 97.77% | 0.8023 | | 78.56% | 0.7292 | |
| Gravel | 67.03% | 0.7693 | 33.801 | 63.13% | 0.7576 | 32.752 | 60.31% | 0.7411 | 30.894 | 61.46% | 0.7857 | 29.773 | 54.17% | 0.7084 | 30.856 |

**Table 6.** Testing classification accuracy and performance metrics.

| Pavia Center Dataset | | | | | | | | | | | | | | | |
|---|---|---|---|---|---|---|---|---|---|---|---|---|---|---|---|
| **Class Name** | **1-D CNN** | | | **2-D CNN** | | | **SC/DN** | | | **RC/DN** | | | **MAME-ZsL** | | |
| | **OA** | **κ** | **McN** | **OA** | **κ** | **McN** | **OA** | **κ** | **McN** | **OA** | **κ** | **McN** | **OA** | **κ** | **McN** |
| Water | 77.83% | 0.8014 | | 79.97% | 0.8208 | | 80.06% | 0.8114 | | 82.77% | 0.8823 | | 62.08% | 0.7539 | |
| Tiles | 81.15% | 0.8296 | 32.587 | 76.72% | 0.7994 | 32.194 | 80.67% | 0.7978 | 31.643 | 78.34% | 0.8095 | 30.588 | 65.37% | 0.7111 | 31.002 |

**Table 7.** The general Pavia dataset.

| Class No | Class Name | All Instances | Sets | | |
|---|---|---|---|---|---|
| | | | **Training** | **Validation** | **Test** |
| 1 | Asphalt | 7995 | √ | - | - |
| 2 | Meadows | 20,013 | √ | - | - |
| 3 | Trees | 4408 | √ | - | - |
| 4 | Bare Soil | 6381 | √ | - | - |
| 5 | Self-Blocking Bricks | 5004 | √ | - | - |
| 6 | Bitumen | 2513 | √ | - | - |
| 7 | Shadows | 1654 | √ | - | - |
| 8 | Metal Sheets | 1610 | - | √ | - |
| 9 | Water | 824 | | √ | - |
| 10 | Gravel | 2491 | - | - | √ |
| 11 | Tiles | 1260 | | - | √ |
| | **Total** | **47,968** | **42,596** | **2434** | **3751** |

As it can be seen from Table 8, the increase of samples has improved the results in the case of the trained algorithms and in the case of the ZsL technique. It is easy to conclude which the proposed MAME-ZsL is a highly valued Deep Learning system which has achieved remarkable results in all evaluations over their respective competing approaches.

The experimental comparison does not include other examples of Zero-shot Learning. This fact does not detract in any case from the value of the proposed method, taking into account that the proposed processing approach builds a predictive model which is comparable to supervised learning systems. The appropriate hyperparameters in the proposed method are also identified by the high reliability and overall accuracy on the obtained results. The final decision was taken based on the performance encountered by the statistical trial and error method. The performance of the proposed model was evaluated against state-of-the-art fully supervised Deep Learning models. It is worth mentioning that the proposed model was trained with instances only from seven classes while the Deep Learning models were trained with instances from all eleven classes. The presented numerical experiments demonstrate that the proposed model produces remarkable results compared to theoretically superior models, providing convincing arguments regarding the classification efficiency of the proposed approach. Another important observation is that it produces accurate results without recurring problems of undetermined cause, because all of the features in the considered dataset are efficiently evaluated. The values of the obtained kappa index are a proof of high reliability (the reliability can be considered as high when $\kappa \geq 0.70$)[35,36].

The superiority of the introduced, novel model focuses on its robustness, accuracy and generalization ability. The overall behavior of the model is comparable to a corresponding supervised one. Specifically, it reduces the possibility of overfitting, it decreases variance or bias and it can fit unseen patterns without reducing its precision. This is a remarkable innovation which significantly improves the overall reliability of the modeling effort. This is the result of the data processing methodology which allows the retention of the more relevant data for upcoming forecasts.

The following Table 8 presents the results obtained by the analysis of the GPD.

The above, Table 8, also provides information on the results of the McNemar test, to assess the importance of the difference between the classification accuracy of the proposed network and the other approaches examined. The improvement of the overall accuracy values obtained by the novel algorithm (compared to the other existing methods) is statistically significant.

Finally, the use of the ensemble approach in this work is related to the fact that very often in multi-factor problems of high complexity such as the one under consideration, the prediction results show multiple variability [43]. This can be attributed to the sensitivity of the correlation models to the data. The main imperative advantage of the proposed ensemble model is the improvement of the overall predictions and the generalization ability (adaptation in new previously unseen data). The ensemble method definitely decreases the overall risk of a particularly poorer choice.

The employed bagging technique offers better prediction and stability, as the overall behavior of the model becomes less noisy and the overall risk of a particularly bad choice that may be caused from under-sampling is significantly reduced. The above assumption is also supported by the dispersion of the expected error, which is close to the mean error value, something which strongly indicates the reliability of the system and the generalization capability that it presents.

As it can be seen in Tables 5–8, the ensemble method appears to have the same or a slightly lower performance across all datasets, compared to the winning (most accurate) algorithm. The highly overall accuracy shows the rate of positive predictions, whereas k reliability index specifies the rate of positive events which were correctly predicted. In all cases, the proposed model has high average accuracy and very high reliability, which means that the ensemble method is a robust and stable approach which returns substantial results. Tables 7 and 8 show clearly that the ensemble model is very promising.

**Table 8.** Testing classification accuracy and performance metrics.

| General Pavia Dataset (OA Is the Overall Accuracy) | | | | | | | | | | | | | | |
| --- | --- | --- | --- | --- | --- | --- | --- | --- | --- | --- | --- | --- | --- | --- |
| **Class Name** | **1-D CNN** | | | **2-D CNN** | | | **SC/DN** | | | **RC/DN** | | | **MAME-ZsL** | | |
| Metal Sheets | **OA** | **κ** | **McN** | **OA** | **κ** | **McN** | **OA** | **κ** | **McN** | **OA** | **κ** | **McN** | **OA** | **κ** | **McN** |
| | 99.44% | 0.8992 | | 100% | 1 | | 99.09% | 0.8734 | | 97.95% | 0.8137 | | 81.16% | 0.8065 | |
| Water | **OA** | **κ** | | **OA** | **κ** | | **OA** | **κ** | | **OA** | **κ** | | **OA** | **κ** | |
| | 81.15% | 0.8296 | | 80.06% | 0.8137 | | 81.82% | 0.8123 | | 84.51% | 0.8119 | | 65.98% | 0.7602 | |
| Gravel | **OA** | **κ** | 30.172 | **OA** | **κ** | 33.847 | **OA** | **κ** | 31.118 | **OA** | **κ** | 30.633 | **OA** | **κ** | 31.647 |
| | 67.97% | 0.7422 | | 64.17% | 0.7393 | | 61.98% | 0.7446 | | 63.98% | 0.7559 | | 54.48% | 0.7021 | |
| Tiles | **OA** | **κ** | | **OA** | **κ** | | **OA** | **κ** | | **OA** | **κ** | | **OA** | **κ** | |
| | 85.11% | 0.8966 | | 80.29% | 0.8420 | | 81.26% | 0.8222 | | 79.96% | 0.8145 | | 69.12% | 0.7452 | |

## 8. Discussion and Conclusions

This research paper proposes a highly effective geographic object-based scene classification system for image *Remote Sensing* which employs a novel ZsL architecture. It introduces serious prerequisites for even more sophisticated pattern recognition systems without prior training.

To the best of our knowledge, it is the first time that such an algorithm has been presented in the literature. It facilitates learning of specialized intermediate representation extraction functions, under complex Deep Learning architectures. Additionally, it utilizes first and second order derivatives as a pre-training method for learning parameters which do not cause exploding or diminishing gradients. Finally, it avoids potential overfitting, while it significantly reduces computational costs and training time. It produces improved training stability, high overall performance and remarkable classification accuracy.

Likewise, the ensemble method used leads to much better prediction results, while providing generalization which is one of the key requirements in the field of machine learning [38]. At the same time, it reduces bias and variance and it eliminates overfitting, by implementing a robust model capable of responding to high complexity problems.

The proposed MAME-ZsL algorithm follows a heuristic hierarchical hyperparameter search methodology, using intermediate representations extracted from the employed neural network and avoiding other irrelevant ones. Using these elements, it discovers appropriate representations which can correctly classify unknown image samples. What should also be emphasized is the use of bootstrap sampling, which accurately addresses noisy scattered misclassification points which other spectral classification methods cannot handle.

The implementation of MAME-ZsL, utilizing the MAML++ algorithm, is based on the optimal use and combination of two highly efficient and fast learning processes (*Softmax* activation function and *AdaBound* algorithm) which create an integrated intelligent system. It is the first time which this hybrid approach has been introduced in the literature.

The successful choice of the *Softmax* (SFM) function instead of the *Sigmoid* (SIG) was based on the fact that it performs better on multi-classification problems, such as the one under consideration, and the sum of its probabilities equals to 1. On the other hand, the *Sigmoid* is used for binary classification tasks. Finally, in the case of SFM, high values have the highest probabilities, whereas in SIG this is not the case.

The use of the AdaBound algorithm offers high convergence speed compared to stochastic gradient descent models. Moreover, it exceedes the poor generalization ability of the adaptive approaches, as it has dynamic limits on the Learning Rate in order to obtain the highest accuracy for the dataset under consideration. Still, this network remarkably implements a GeoAI approach for large-scale geospatial data analysis which attempts to balance latency, throughput and fault-tolerance using ZsL. At the same time, it makes effective use of the intermediate representations of Deep Learning.

The proposed model avoids overfitting, decreases variance or bias, and it can fit unseen patterns, without reducing its performance.

The reliability of the proposed network has been proven in identifying scenes from Remote Sensing photographs. This suggests that it can be used in higher level geospatial data analysis processes, such as multi-sector classification, recognition and monitoring of specific patterns and sensors' data fusion. In addition, the performance of the proposed model was evaluated against state-of-the-art supervised Deep Learning models. It is worth mentioning that the proposed model was trained with instances only from seven classes, while the other models were trained with instances from all eleven classes. The presented numerical experiments demonstrate that the introduced approach produces remarkable results compared to theoretically superior models, providing convincing arguments regarding its classification efficiency.

Suggestions for the evolution and future improvements of this network should focus on comparison with other ZsL models. It is interesting to see the difference between ZsL, one-shot and five-shot learning methods, in terms of efficiency.

On the other hand, future research could focus on further optimization of the hyperparameters of the algorithms used in the proposed MAME-ZsL architecture. This may lead to an even more efficient, more accurate and faster classification process, either by using a heuristic approach or by employing a potential adjustment of the algorithm with spiking neural networks [44].

Additionally, it would be important to study the extension of this system by implementing more complex architectures with *Siamese neural networks* in parallel and distributed real time data stream environments [45].

Finally, an additional element which could be considered in the direction of future expansion, concerns the operation of the network by means of self-improvement and redefinition of its parameters automatically. It will thus be able to fully automate the process of extracting useful intermediate representations from Deep Learning techniques.

**Author Contributions:** Conceptualization, K.D. and L.I.; investigation, K.D.; methodology, K.D. and L.I.; software, K.D. and L.I.; validation, K.D. and L.I.; formal analysis L.I.; resources, K.D. and L.I.; data curation, K.D. and L.I.; writing—original draft preparation, K.D.; writing—review and editing L.I.; supervision, L.I. All authors have read and agreed to the published version of the manuscript.

**Funding:** This research received no external funding.

**Conflicts of Interest:** The authors declare no conflict of interest.

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
