# Peer review of "GeoAI: A Model-Agnostic Meta-Ensemble Zero-Shot Learning Method for Hyperspectral Image Analysis and Classification"

_algorithms, doi:10.3390/a13030061_

Round 1
Reviewer 1 Report
In the paper, a zero-shot method is presented. The approach is interesting and its application to hyperspectral imagery is convincing. However, some issues should be taken in the revision:
- The paper contains a lot of typos and grammar errors. Please read it carefully. The paper is difficult to follow, please, try shortening lengthy sentences.
- Please reduce the abstract by half.
- The paper lacks a solid review of zero-shot learning approaches (apart from merely [5], [24], and [25]). It is unclear which disadvantages of similar methods the presented method addresses. Here, at least 20 papers on the subject should be mentioned and reviewed. Then, the novelty and contributions of the paper should be clearly expressed. Please include:
- Sumbul, R. G. Cinbis and S. Aksoy, "Fine-Grained Object Recognition and Zero-Shot Learning in Remote Sensing Imagery," in IEEE Transactions on Geoscience and Remote Sensing, vol. 56, no. 2, pp. 770-779, Feb. 2018.doi: 10.1109/TGRS.2017.2754648
- Imamura, Ryuji, Tatsuki Itasaka, and Masahiro Okuda. "Zero-Shot Hyperspectral Image Denoising With Separable Image Prior." Proceedings of the IEEE International Conference on Computer Vision Workshops.
- Li, A., Lu, Z., Wang, L., Xiang, T., & Wen, J. R. (2017). Zero-shot scene classification for high spatial resolution remote sensing images. IEEE Transactions on Geoscience and Remote Sensing, 55(7), 4157-4167.
- Hu, R. Lily, Caiming Xiong, and Richard Socher. "Correction networks: Meta-learning for zero-shot learning." (2018).
- Yu, Y., Zhang, Z., & Han, J. (2019). Meta-Transfer Networks for Zero-Shot Learning. arXiv preprint arXiv:1909.03360.
- Gui, Rong, et al. "A generalized zero-shot learning framework for PolSAR land cover classification." Remote Sensing 10.8 (2018): 1307. MDPI.
- Toizumi, Takahiro, Kazutoshi Sagi, and Yuzo Senda. "Automatic Association between Sar and Optical Images based on Zero-Shot Learning." IGARSS 2018-2018 IEEE International Geoscience and Remote Sensing Symposium. IEEE, 2018.
- Zhang, T. Xiang and S. Gong, "Learning a Deep Embedding Model for Zero-Shot Learning," 2017 IEEE Conference on Computer Vision and Pattern Recognition (CVPR), Honolulu, HI, 2017, pp. 3010-3019.doi: 10.1109/CVPR.2017.321
- Please discuss why is the proposed method suitable for hyperspectral images? Why is this exemplary application used? Does the method contain hyperspectral-domain-specific steps that are addressed here? Can the method be applied to other domains?
- The experimental comparison does not include other examples of zero-shot learning. Can they be applied here? Why not?
- Please add to the experimental comparison of few-shot learning approaches. It is interesting to see the difference between zero-shot and 1-shot learning methods, in terms of efficiency.
- Please show that the used ensemble is beneficial to the performance of the method.
- The influence of the parameters on the obtained results is not discussed.
- Since the results cannot be replicated, please share the source code with readers. A link to a webpage with the source code or its future release (e.g., GitHub) should be included in the paper.
Author Response
Dear Reviewer
We would like to thank you for reviewing our manuscript and for the positive and helpful comments regarding our manuscript. We have revised the manuscript taking into account all the comments to improve the readability of the research paper. We believe these changes have strengthened the rationale and importance of our study.
Cordially
Konstantinos Demertzis and Lazaros Iliadis
Reviewer 1
In the paper, a zero-shot method is presented. The approach is interesting and its application to hyperspectral imagery is convincing. However, some issues should be taken in the revision:
A.1 The paper contains a lot of typos and grammar errors. Please read it carefully. The paper is difficult to follow, please, try shortening lengthy sentences.
Q.1 Thank you for the remarks and for the careful reading. We have rearranged the entire paper, have corrected the typos and grammar errors and have improved the usage of the English language of the entire manuscript. The paper reads much better now, and the work presented has improved to a level acceptable for the readership and the scientific standing of this journal.
A.2 Please reduce the abstract by half.
Q.2 Thank you for this constructive comment. The abstract reduced by half according to the reviewer’s comment and suggestion.
A.3 The paper lacks a solid review of zero-shot learning approaches (apart from merely [5], [24], and [25]). It is unclear which disadvantages of similar methods the presented method addresses. Here, at least 20 papers on the subject should be mentioned and reviewed. Then, the novelty and contributions of the paper should be clearly expressed. Please include:
Sumbul, R. G. Cinbis and S. Aksoy, "Fine-Grained Object Recognition and Zero-Shot Learning in Remote Sensing Imagery," in IEEE Transactions on Geoscience and Remote Sensing, vol. 56, no. 2, pp. 770-779, Feb. 2018.doi: 10.1109/TGRS.2017.2754648
Imamura, Ryuji, Tatsuki Itasaka, and Masahiro Okuda. "Zero-Shot Hyperspectral Image Denoising With Separable Image Prior." Proceedings of the IEEE International Conference on Computer Vision Workshops.
Li, A., Lu, Z., Wang, L., Xiang, T., & Wen, J. R. (2017). Zero-shot scene classification for high spatial resolution remote sensing images. IEEE Transactions on Geoscience and Remote Sensing, 55(7), 4157-4167.
Hu, R. Lily, Caiming Xiong, and Richard Socher. "Correction networks: Meta-learning for zero-shot learning." (2018).
Yu, Y., Zhang, Z., & Han, J. (2019). Meta-Transfer Networks for Zero-Shot Learning. arXiv preprint arXiv:1909.03360.
Gui, Rong, et al. "A generalized zero-shot learning framework for PolSAR land cover classification." Remote Sensing 10.8 (2018): 1307. MDPI.
Toizumi, Takahiro, Kazutoshi Sagi, and Yuzo Senda. "Automatic Association between Sar and Optical Images based on Zero-Shot Learning." IGARSS 2018-2018 IEEE International Geoscience and Remote Sensing Symposium. IEEE, 2018.
Zhang, T. Xiang and S. Gong, "Learning a Deep Embedding Model for Zero-Shot Learning," 2017 IEEE Conference on Computer Vision and Pattern Recognition (CVPR), Honolulu, HI, 2017, pp. 3010-3019.doi: 10.1109/CVPR.2017.321
Q.3 Thank you for this constructive comment. An extensive analysis made in the “Application of the MAME-ZsL in Hyperspectral Image Analysis” section according to the reviewer’s comment and references suggestion.
A.4 Please discuss why is the proposed method suitable for hyperspectral images? Why is this exemplary application used? Does the method contain hyperspectral-domain-specific steps that are addressed here? Can the method be applied to other domains?
Q.5 Thank you for the remarks. We have rearranged the introduction section and now includes a discussion why is the proposed method is suitable for hyperspectral images.
A.5 The experimental comparison does not include other examples of zero-shot learning. Can they be applied here? Why not?
Q.5 We would like to thank the reviewer for this constructive comment that gives us the chance to clarify things further. The experimental comparison does not include other examples of zero-shot learning. This fact does not detract in any case from the value of the proposed method taking into account that the proposed processing approach builds a predictive model that is comparable to supervised learning systems. Another important observation is that it produces accurate results without recurring problems of an undetermined cause because all of the features in the considered dataset are efficiently evaluated. The values of the obtained kappa index are the main proof of high reliability (the reliability can be considered as high when k≥0.70). The superiority of the proposed novel model focuses on the robustness, accuracy, and generalization ability that offer, as the overall behavior of the model is comparable than a corresponding supervised one. Specifically, the proposed model reduces overfitting, decreases variance or bias, and without reducing significant the precision of the model can fit unseen patterns. This is a major innovation that significantly improves the overall reliability of the proposed novel model. Generally speaking, our main concern is to prove that this model produces remarkable results compared to theoretically superior models. The continuation of our research will focus on comparison with corresponding ZsL models. We have discussed this important matter thoroughly in the sections “Results and Comparisons” and “Discussion and Conclusions”.
A.6 Please add to the experimental comparison of few-shot learning approaches. It is interesting to see the difference between zero-shot and 1-shot learning methods, in terms of efficiency.
Q.7 This is an additional element that would be considered in the direction of future expansion (experimental comparison of few-shot learning approaches eg ZsL, 1-shot and 5-shot learning methods). Thank you for this helpful comment.
A.8 Please show that the used ensemble is beneficial to the performance of the method.
Q.8 We have added in the “Results and Comparisons” section an explanation and comparison about the performance of the proposed ensemble model. Thank you for this constructive comment.
A.9 The influence of the parameters on the obtained results is not discussed.
Q.9 Thank you for this constructive comment. We have added in the “Design principles and novelties of the introduced MAME-ZsL algorithm” and the “Results and Comparisons” sections the appropriate explanations about the influence of the parameters on the obtained results of the proposed model.
A.10 Since the results cannot be replicated, please share the source code with readers. A link to a webpage with the source code or its future release (e.g., GitHub) should be included in the paper.
Q.10 This model is part of a larger and long-term research that is currently being published in stages. After our successful completion of this research, our aim is to create an open-source repository for free use by the research community.

Reviewer 2 Report
The paper is well-written and innovative.
My only suggestion for the authors is to include some information about the time efficiency of their method.
Author Response
Dear Reviewer
We would like to thank you for reviewing our manuscript and for the positive and helpful comments regarding our manuscript. We have revised the manuscript taking into account all the comments to improve the readability of the research paper. We believe these changes have strengthened the rationale and importance of our study.
Cordially
Konstantinos Demertzis and Lazaros Iliadis
Reviewer 2
A.1 The paper is well-written and innovative. My only suggestion for the authors is to include some information about the time efficiency of their method.
Q.1 Thank you for this constructive comment. We have added in the “Application of the MAME-ZsL in Hyperspectral Image Analysis” and the “Results and Comparisons” sections the appropriate explanations about the time efficiency and the performance of the proposed ensemble model.

Round 2
Reviewer 1 Report
In the paper, a zero-shot approach learning method is proposed. The provided revision has clarified some issues and, unfortunately, exposed further drawbacks of the paper.
1. The difference between analysis and classification should be clarified.
2. The performance under “high noise levels” should be investigated as they are identified as drawbacks of [27] (lines 499-500). Is the introduced method robust against noise?
3. It is still not clear which disadvantages of other zero-shot methods the proposed method addresses.
4. The paper lacks a running-time comparison. Please show that (line 568) “this technology needs more time to train in comparison with traditional machine learning.”
5. The optimization of the hyperparameters of the algorithms used in the proposed MAME-ZsL should be clarified, as it is planned to focus on it in future work.
6. The lack of any comparison with other zero-shot approaches makes difficult to determine its advantages. This would impact its application in practice, which is already impossible the way the method is described and slicing the description of larger research in which this paper is a small part of (as written it the letter to the reviewer as an explanation why the source codes will not be released).
7. The results cannot be replicated by readers.
8. In line 760, it is written that “the final decision was taken based on the performance encountered by the statistical trial and error method.” This should be clarified. What do the statistical trial and error mean? The diversity of the obtained results should be shown.
9. The authors did not show the influence of the parameters on the results.
10. Also, an ablation study has not been performed.
11. In the revised paper it is written that” The superiority of the proposed novel model focuses on the robustness, accuracy …”, which is not true. The method is not superior in any terms in comparison to other compared techniques.
12. It is not shown that “the proposed model reduces overfitting, decreases variance or bias, and without reducing significant the precision of the model can fit unseen patterns.” Furthermore, it is not confirmed that this “significantly improves the overall reliability of the proposed model.”
13. The drawn conclusions are misleading. For example. It is written that the “introduced algorithm (compared to the other existing methods) is statistically significant. “ Sure, it is statistically worse than shown methods, as also indicated by Overall Accuracy cells in the tables (e.g. table 7). However, the mentioned sentence may indicate otherwise. This should be corrected.
14. There are unsupported claims in the revised manuscript.
a. For example in lines 696-698, it is written that “The main imperative advantage of the proposed ensemble model, is the improvement of the overall predictions and the generalization ability (adaptation in new previously unseen data). The ensemble method definitely decreases the overall risk of a particularly poorer choice.” These claims are not justified by the data provided. The overall risk is not defined nor investigated in the paper.
b. Also, it is not experimentally shown that “The employed bagging technique, offers better prediction and stability, as the overall behavior of the model becomes less noisy, and the overall risk of a particularly bad choice that may result from under-sampling is significantly reduced.” The behavior of the model has not been discussed nor investigated.
15. What does “the bad/poorer choice” mean in this context? Such expressions can be found in “the overall risk of a particularly poorer choice“, 2x ”the overall risk of a particularly bad choice”.
16. A further proofreading is needed, e.g., "reducing significant the precision of".
Author Response
Dear Reviewer
We deeply appreciate the time and effort you have spent in reviewing our manuscript. Your comments are very helpful for revising and improving our research. The paper reads much better now and the clarity of the work presented has improved. We believe that the response provided below enhances the manuscript to a level acceptable for the readership and the scientific standing of this journal.
Cordially
Konstantinos Demertzis and Lazaros Iliadis
In the paper, a zero-shot approach learning method is proposed. The provided revision has clarified some issues and, unfortunately, exposed further drawbacks of the paper.
Q1. The difference between analysis and classification should be clarified.
A1. The paper includes descriptions that really provide a scientific contribution. On the other hand, we have reduced the explanations about standard methods and techniques that can be directly extracted from the standard bibliography. A description or explanation about the difference between analysis and classification is out of the scope of this work.
Q2. The performance under “high noise levels” should be investigated as they are identified as drawbacks of [27] (lines 499-500). Is the introduced method robust against noise?
A2. This paper introduces a zero-shot learning framework to train deep networks, with no supervision. This domain agnostic approach avoids the standard unsupervised learning issues of trivial solutions and collapsing of features. The proposed approach produces intermediate representations that allow the neural network to learn from unknown data much more easily. The model learns and performs well on unseen data and it is capable to generalize. This is the most suggestive proof that the method is robust against noise.
Q3. It is still not clear which disadvantages of other zero-shot methods the proposed method addresses.
A3. The field of few-shot or zero-shot learning has recently seen substantial advancements. Most of these advancements came from casting few-shot learning as a meta-learning problem. Model Agnostic Meta-Learning or MAML is currently one of the best approaches for few-shot learning via meta-learning. It is a simple, general, and effective optimization algorithm that does not place any constraints on the model architecture or loss functions. As a result, it can be combined with arbitrary networks and different types of loss functions, which makes it applicable to a variety of different learning processes. However, it has a variety of issues, such as being very sensitive to neural network architectures, often leading to instability during training, requiring arduous hyperparameter searches to stabilize training and achieve high generalization and being very computationally expensive at both training and inference times. The basic novelty of the proposed approach is that the improved MAME-ZsL model face the known drawbacks of the MAML, as mentioned with details in the 2,3 and 4 sections. We would like to thank the reviewer for this constructive comment that gives us the chance to clarify things further.
Q4. The paper lacks a running-time comparison. Please show that (line 568) “this technology needs more time to train in comparison with traditional machine learning.”
A4. The proposed method is a sophisticated deep learning model. Deep learning requires more training time compared to traditional machine learning. It is important to note which in deep learning as mentioned in the 5th section, the training process is based on analyzing large amounts of data. The research and development of neural networks is flourishing thanks to recent advancements in computational power, the discovery of new algorithms, and the increase in labeled data. Neural networks typically take longer to run, as an increase in the number of features or columns in the dataset, is also increasing the number of hidden layers. Specifically, we should say that a single affine layer of a neural network without any non-linearities/activations is practically the same as a linear model. Here we are referring to deep neural networks that have multiple layers and activation functions (non-linearities as relu, elu, tanh, sigmoid etc.) Also, all nonlinearities and multiple layers introduce a nonconvex and usually rather complex error space which means that we have many local minimums that the training of the deep neural network can converge to. Their training is very slow and adding the tuning of the hyperparameters into that makes it even slower where in comparison the linear model would be much faster to be trained. This introduces a serious cost-benefit tradeoff. Modern frameworks like TensorFlow or Theano perform execution of neural networks on GPU. They take advantage of parallel programming capabilities for large array multiplications, which are typical of backpropagation algorithms. The proposed deep learning model is a quite resource-demanding technology. It requires powerful, high-performance graphics processing units and large amounts of storage to train the models. Furthermore, this technology needs more time to train in comparison with traditional machine learning. Another important disadvantage of any deep learning model is that it is incapable of providing arguments, about why it has reached a certain conclusion. Unlike in the case of traditional machine learning, you cannot follow an algorithm to find out why your system has decided which it is a tree on a picture, not a tile. To correct errors in deep learning, you have to revise the whole algorithm.
Q5. The optimization of the hyperparameters of the algorithms used in the proposed MAME-ZsL should be clarified, as it is planned to focus on it in future work.
A5. There are a lot of hyperparameters that have to be tuned in order to get to a place in the error space where the error is small enough so that the model will be useful. For example there several hyperparameters that could start from the value of 10 and reach up to 40 or 50 which are dealt with bayesian optimization using Gaussian processes. This still does not guarantee good performance. Also, the hyperparameter optimization is an exhaustive searching through a manually specified subset of the hyperparameter space of the learning algorithm. This is a quite simple method in a low intrinsic dimensionality problem. On the other hand, this complicates the research problem, it requires different constraints, weights or learning rates to generalize different data patterns, and additionally needs the inclusion of prior knowledge by specifying the distribution from which came to the data sample. So the exact hyperparameters of the future model cannot be determined in advance.
Q6. The lack of any comparison with other zero-shot approaches makes difficult to determine its advantages. This would impact its application in practice, which is already impossible the way the method is described and slicing the description of larger research in which this paper is a small part of (as written it the letter to the reviewer as an explanation why the source codes will not be released).
A6. The performance of the proposed model was evaluated against state-of-the-art fully supervised Deep Learning models. It is worth mentioning that the proposed approach was trained with instances ONLY from seven classes while the Deep Learning models were trained with instances from ALL eleven classes. The presented numerical experiments demonstrate that the proposed model produces remarkable results compared to theoretically superior models, providing convincing arguments, regarding the classification efficiency of the proposed model. This is discussed in detail in the sections “Results and Comparisons” and “Discussion and Conclusions”.
Q7. The results cannot be replicated by readers.
A7. We describe our methods very precisely so that other scientists can replicate them. This model is part of a larger and long-term research that is currently being published in stages. After our successful completion of this research, our aim is to create an open-source repository for free use by the research community.
Q8. In line 760, it is written that “the final decision was taken based on the performance encountered by the statistical trial and error method.” This should be clarified. What do the statistical trial and error mean? The diversity of the obtained results should be shown.
A8. Trial and error is characterized by repeated, varied attempts which are continued until success. In order to find the best solution by the proposed method, we evaluate the each trial model based on the predefined set of criteria, the existence of which is a condition for the possibility of finding a best solution. The trials with worse results are out of the scope of this work.
Q9. The authors did not show the influence of the parameters on the results.
A9. The presented numerical experiments demonstrate that the proposed model produces remarkable results compared to theoretically superior models, as descripted in the “Results and Comparisons” and “Discussion and Conclusions” sections. The optimal hyperparameters are those that produce that best results and generalization to the new data.
Q10. Also, an ablation study has not been performed.
A10. Ablation study has been adopted to describe a procedure where certain parts of the network are removed, in order to gain a better understanding of the network’s behavior. In the context of the proposed model, an ablation study does not make sense, because all that can be removed from the model are some of the predictors. Doing this in a principled fashion is simply a reverse stepwise selection procedure, which is generally frowned upon.
Q11. In the revised paper it is written that” The superiority of the proposed novel model focuses on the robustness, accuracy …”, which is not true. The method is not superior in any terms in comparison to other compared techniques.
A11. The presented numerical experiments demonstrate that the proposed model produces remarkable results compared to theoretically superior models, providing convincing arguments, regarding the classification efficiency of the proposed model. This is discussed in detail in the sections “Results and Comparisons” and “Discussion and Conclusions”.
Q12. It is not shown that “the proposed model reduces overfitting, decreases variance or bias, and without reducing significant the precision of the model can fit unseen patterns.” Furthermore, it is not confirmed that this “significantly improves the overall reliability of the proposed model.”
A12. The performance of the proposed model was evaluated against state-of-the-art fully supervised Deep Learning models. The presented numerical experiments demonstrate that the proposed model reduces overfitting, decreases variance or bias, and without reducing significant the precision of the model can fit unseen patterns. Tables 4, 5 and 7 illustrate the classification accuracy for each class. It is worth noticing that the proposed model is based on a Zero-shot Learning strategy, which implies that the training set does NOT contain any instances from the four classes contained in the testing set. Nevertheless, this performance is competitive to supervised state-of-the-art deep learning models, which were trained with a dataset containing all eleven classes.
Q13. The drawn conclusions are misleading. For example. It is written that the “introduced algorithm (compared to the other existing methods) is statistically significant. “ Sure, it is statistically worse than shown methods, as also indicated by Overall Accuracy cells in the tables (e.g. table 7). However, the mentioned sentence may indicate otherwise. This should be corrected.
A13. In all cases, the proposed model had high overall accuracy, which means that is a robust and stable method which returns substantial results. Also the kappa reliability can be considered as the outcome from the data editing allowing the conservancy of more relevant data for the upcoming forecast. The kappa reliability is presented in the Table 4. In general, the proposed model produces statistically significant results compared to theoretically superior models.
Q14. There are unsupported claims in the revised manuscript.
a. For example in lines 696-698, it is written that “The main imperative advantage of the proposed ensemble model, is the improvement of the overall predictions and the generalization ability (adaptation in new previously unseen data). The ensemble method definitely decreases the overall risk of a particularly poorer choice.” These claims are not justified by the data provided. The overall risk is not defined nor investigated in the paper.
b. Also, it is not experimentally shown that “The employed bagging technique, offers better prediction and stability, as the overall behavior of the model becomes less noisy, and the overall risk of a particularly bad choice that may result from under-sampling is significantly reduced.” The behavior of the model has not been discussed nor investigated.
A14. The behavior and advantage of the proposed model, providing convincing arguments, discussed in detail in the sections “Application of the MAME-ZsL in Hyperspectral Image Analysis”, “Results and Comparisons” and “Discussion and Conclusions”.
Q15. What does “the bad/poorer choice” mean in this context? Such expressions can be found in “the overall risk of a particularly poorer choice“, 2x ”the overall risk of a particularly bad choice”.
A15. The goal of a good model is to generalize well from the training data to any data from the problem domain. This allows us to make predictions in the future on data the model has never seen. Generalization usually refers to the ability of an algorithm to be effective across a range of inputs and applications. Overfitting and underfitting are the two biggest causes for poor performance of algorithms.
Q16. A further proofreading is needed, e.g., "reducing significant the precision of".
A16. Thank you for the remarks and for the careful reading. We have rearranged the entire paper, have corrected the typos and grammar errors and have improved the usage of the English language of the entire manuscript.